# The Effect of Labor Flexibility on Financial Performance in Korea: The Moderating Effect of Labor Relations Climate

**Hyunmin Choe [1], Yongwon Kim [2] and Sungok Moon [3,*]**

[1] Office of the Senior Vice President for Research, Korea Development Institute, Sejong 30149, Korea; hmchoe@kdi.re.kr
[2] Korea Expressway Corp Research Institute, Hwaseong 18489, Korea; ywkim@ex.co.kr
[3] School of Business, Yonsei University, Seoul 03722, Korea
[*] Correspondence: msok@yonsei.ac.kr; Tel.: +82-2-2123-5469

**Abstract:** This study reveals the impact of labor flexibility (i.e., numerical flexibility, functional flexibility, financial flexibility, and time flexibility) on financial performance and the effect of labor relations climate that moderates the two. Numerical flexibility, functional flexibility, financial flexibility, and time flexibility were chosen as the independent variables, and firms' net profit was selected as the dependent variable to test the hypotheses. Statistical analysis was conducted on 1482 workplaces, and the findings of the regression analysis are as follows. First, numerical flexibility and financial flexibility had a positive effect on net profit among different types of labor flexibility. Second, the labor relations climate had a positive moderating effect on numerical and financial flexibility. This study is critical because it individually tested the four types of labor flexibility and empirically studied the relationship between each type and outcome variables. A labor relations climate strengthens the relationship between labor flexibility and net profit. Therefore, in order to increase the net profit of a company, it is necessary to utilize appropriate numerical and financial flexibility, and it is important to create a good labor–management partnership.

**Keywords:** labor flexibility; labor relations climate; net profit; workplace panel survey

## 1. Introduction

Over the last 20 years, the flexible labor market and job type have been highlighted as crucial topics. All Human Resource Management (HRM) practices have emphasized the capacity of organizations to promptly address technological changes and consumer needs and cope with the rapidly changing environment. Accordingly, flexibility is becoming increasingly important for quick adaptation mismatches in the labor market, price competition in the production market, and financial restructuring in the capital market. In a comprehensive evaluation concerning labor-market flexibility, Korea fell from the 38th place out of 107 countries in 2008 to the 70th position in 2013 [1]. Additionally, Korea's economic growth rate has continuously declined since 2010, and the growth rate in 2020 is $-0.9\%$, showing a low growth rate. In addition, the economic uncertainty that companies are facing has not been resolved. According to the report of the World Economic Forum (WEF), Korea's labor market rigidity ranks 76th, which is seven steps lower than last year. The reason that labor flexibility is important is that effective use of labor force is necessary for the sustainable survival of an organization.

Labor flexibility is a method typically used to overcome emerging labor market problems, such as an aging population caused by the extension of the average lifespan and low birth rates, rise of dual-earner couples, reduction in jobs, and increase in youth unemployment. Additionally, labor flexibility is necessary to maintain competitiveness and achieve sustainable growth in a rapidly changing environment. This is because, in order to satisfy the rapidly changing needs of customers, labor flexibility needs to be effective, and if the labor flexibility can be maintained continuously, the competitive advantage is sustainable.

This paper reviews various types of labor flexibilization that can resolve the pending issues of wage and labor market rigidity currently experienced in Korea, in tandem with the moderating factors that can increase the labor flexibility effect.

Discussions on labor flexibility began in the 1970s, and various studies have been conducted both in Korea and overseas, starting with Atkinson [2]. Previous literature mostly constituted comparative studies among nations, policy studies, and normative studies examining the relevance between flexibility strategies, while few were empirical studies [3–5]. Moreover, some progress has been made on the research on numerical flexibility and functional flexibility, but a multilateral review on various forms of flexibility is lacking [6]. Therefore, this research clarifies each type of labor flexibility and examines how each type affects the financial performance of firms.

Labor-management relation is a key variable that determines the utility and success of a system that a firm intends to adopt and manage [7]. In many cases, however, systems are implemented without reaching a labor–management agreement in the flexibilization of the labor market, causing labor–management conflicts and creating obstacles for labor flexibility strategies. There could be some effect if labor relations are in an amicable climate where consensus and cooperation of workers can be anticipated. Even though labor relations climate is critical for the implementation of labor flexibility, there is insufficient empirical research. Thus, this study verifies the moderating effect of a labor relations climate.

In this study, we will classify the subconstructs of labor flexibility and examine the effect of each type of labor flexibility on net profit. Moreover, from a contingency perspective, this study aims to demonstrate the effect of the labor relations climate moderating the relationship between labor flexibility and net profit.

## 2. Theoretical Background and Hypotheses

### 2.1. Labor Flexibility and Firm Performance

Whyman and Petrescu [8] defined labor flexibility as work arrangements and patterns aimed to enable employees and employers to adjust corporate activities in order to adapt to the demands of the working life and the economic climate. It refers to the firms' ability to quickly and efficiently allocate human resources to address the changes in quantitative and qualitative demands brought by fluctuating production requirements. This concept enables firms to flexibly supply human resources in a changing economic environment and revitalize the economy by promoting functional efficiency and flexibility of employment, wages, and workers, ultimately reducing unemployment.

Furthermore, comprehensive theories on firms' labor flexibility are difficult to find but can be explained by the transaction cost theory [9] and the resource-based theory [10]. These theories claim that employment flexibility refers to the employment of external labor markets, existing outside the firm's boundaries, and explain the reason for its use. According to the transaction cost theory, costs can be decreased more through the employment of temporary workers belonging to the external labor market than through the rigid internal labor market. Internally securing a core group is vital in the resource-based theory. As argued by Atkinson [2], an easily replaceable noncore group does not have to be secured and retained internally, which is why there is employment externalization.

Different researchers have presented various classifications for labor flexibility. After introducing the concept of flexibility, Atkinson [2] mentioned that this notion is unclear and further classified it into numerical flexibility, functional flexibility, and financial flexibility. McIlroy et al. [11] classify the concept of numerical flexibility into external numerical flexibility, internal numerical flexibility, functional flexibility, and financial flexibility. Budd [12] categorized flexibility into employment flexibility (i.e., adjusting labor utilization by diversifying working hours and the number of workers), wage flexibility (i.e., establishing a reward system suitable for competition and organizational performance), functional flexibility (i.e., easily reshuffling workers to other jobs depending on customer demand and production need), and procedural flexibility (i.e., changing the production method, technology, work system, and composition). Accordingly, labor market flexibility is sorted

based on various criteria but is generally categorized into four types, namely numerical flexibility, functional flexibility, financial flexibility, and time flexibility [13].

The first type of labor flexibility is numerical flexibility. Using numerical flexibility has a positive effect on firms' financial performance. The utilization of temporary workers and external labor resources has a positive effect because the introduction and accumulation of new knowledge within the organization [14] enables innovation [15]. In turn, the creativity and innovation of new workers contribute to positive changes in the organization [16]. Moreover, fixed costs related to employee retention can be reduced by using externalization, which is known to have a positive effect on improving the performance and productivity of workers, ultimately producing financial outcomes and sustainable growth [17] for the firm [18,19]. Therefore, the following hypothesis was set up.

**Hypothesis 1a.** *Numerical flexibility is positively related to net profit.*

Functional flexibility or internal flexibility is about adjusting employment without reducing the number of workers. This type adjusts the amount of labor or flexibly uses an organization's internal staff. The functional flexibility strategy diversifies products to more easily adjust to changing market conditions and increases workers' adaptability to new technologies. This concept includes all methods of involving workers in decision making and reinforcing capacity building, as well as job rotation, multi-functionalization, multi-skilled staff training, relocation, labor mobility among workplaces, and transfer. The most typical example to obtain diverse work experiences for workers is a job rotation. The objective of functional flexibility is to improve performance by increasing the competencies of internal workers, together with their organizational and job compatibility, enabling them to admirably perform any job or task given to them within the firm [20]. Systems, such as those staffing the right person in the right place and using the internal job market, impact organizational performance [21], and investment in employee socialization, training, information sharing, and job analysis develops individual abilities, exposing workers to diverse experiences, knowledge, and functions [22,23]. Hence, workers are able to develop more complex technologies, take new risks, and strive to improve their job competencies [14]. Their improved competencies would result in better firm performance and sustainable growth. Therefore, the following hypothesis was considered.

**Hypothesis 1b.** *Functional flexibility is positively related to net profit.*

Financial flexibility or pay flexibility (or wage flexibility) is a wage determination method aligned with organizational performance and firm productivity. This type signifies a performance-related pay system aligned with individual or team performance or the wage peak system, which was further aligned with productivity. Workers who receive wages based on performance and productivity will endeavor to improve their productivity, leading to better firm performance and sustainable growth.

Practices used in financial flexibility also seek suitable rewards in relation to individual performance, such as bonuses and pay-for-performance, which raises trust and competition, leading to better overall performance [24]. Many studies have revealed a relationship between the adoption of a pay-for-performance system and business performance [25,26].

The expectancy theory [27] and goal-setting theory [28] explain that performance-related pay can motivate workers. The implementation of the performance-related pay system increases the awareness of procedural justice, instilling the idea that corporate profits are not just distributed to specific people, such as managers or shareholders, but are also shared with workers, thereby increasing trust in the management as well as organizational commitment [29,30]. Accordingly, the following hypothesis is set with regard to financial flexibility.

**Hypothesis 1c.** *Financial flexibility is positively related to net profit.*

Time flexibility specifies the adjustment of working hours without the adjustment of the number of workers working within the firm. Time flexibility is a management strategy that increases productivity and provides flexibility to an organization through various working systems that fulfill the conditions and characteristics of the organization. The importance of this type is more emphasized via social changes, such as advancement in information and communications technology and the increase in women in the paid workforce. Time flexibility comprises core time work, selective working hours, shift work, extended working hours, flexible working hours, mobile office, working from home, and so on. [13]. Many studies have been conducted on work systems with flexible hours, proving various effects such as motivating workers, improving production performance and quality, enhancing organizational performance [31,32], boosting workers' job satisfaction [33], decreasing absenteeism [34] and improving work and life balance [35]. Workers may feel that they are being well treated, with care and consideration, if working hours and schedules are adjusted flexibly. Therefore, the following hypothesis can be assumed.

**Hypothesis 1d.** *Time flexibility is positively related to net profit.*

### 2.2. Moderating Effects of Labor Relations Climate

For a firm to adopt and use a certain system and achieve a particular goal, a relationship with workers as members of the organization is very important, which is why labor flexibilization strategies are closely interlinked with labor relations. Cooperative labor relations serve as an antecedent determining job satisfaction or organizational involvement [7]. Studies are actively conducted on the labor–management partnership, which is an attribute of cooperative labor relations. Various concepts are used by different researchers, but mostly it refers to the type of labor relations, whereby workers can be involved in management [36]. According to the model presented by Deery and Iverson [7], labor–management relations were influenced by information sharing and seamless communication, as well as union executives' labor relations strategies. Labor relations based on cooperation and trust serve as a core cognitive attribute of labor-management partnership, encouraging long-term and multidimensional participation of workers.

Labor–management relations have a critical effect on organizational performance [37]. Specifically, the overall climate perceived in labor relations directly or indirectly affects the efficiency of organizational level production performance [7]. A cooperative and amicable labor relations climate improves corporate performance [38], enhances productivity, and decreases absenteeism [7]. Thus, how workers in a firm using a flexibility strategy perceive the overall labor relations climate will impact how the labor flexibility strategy leads to increased firm's financial performance.

Numerical flexibility has a negative impact on the employment security [39]. However, if the labor relation climate is favorable, workers will feel less job insecurity. A good labor relation climate will not only mitigate the negative effects, but also the numerical flexibility agreed upon between labor and management will make the company expect it to be a driving force to exit crisis or achieve better performance. Therefore, labor relation climate will positively moderate the relationship between numerical flexibility and financial performance.

Learning various tasks is an important career path that increases the employability of workers, but excessive flexibility causes overwork [23]. However, if the labor relation climate is favorable, workers can develop their careers and expect better performance and rewards through the functional flexibility. Therefore, labor relation climate will positively moderate the relationship between functional flexibility and financial performance.

The positive effect of the pay-for-performance system on the motivation of employees is reinforced when labor–management relations are good. This is because, in the case of pay-for-performance, fair evaluation must precede, and if labor relation climate is favorable, employees will trust that the performance evaluation is done fairly. Therefore,

labor relation climate will positively moderate the relationship between financial flexibility and financial performance.

Time flexibility is related to work autonomy [40]. When the labor relation climate is favorable, the motivation and commitment will increase because the given autonomy is genuinely felt. In addition, workers will feel that support and treatment for them will increase, so they will become more committed to their work. Therefore, labor relation climate will positively moderate the relationship between time flexibility and financial performance. Therefore, the following hypothesis was set up (See Figure 1).

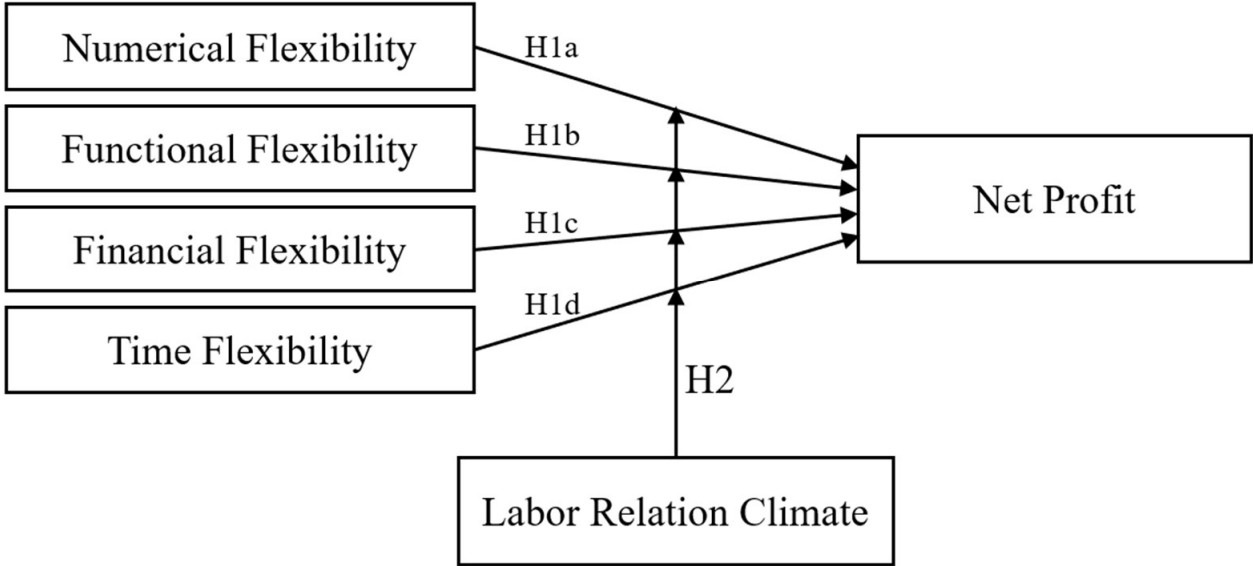

**Figure 1.** The theoretical model of the research.

**Hypothesis 2.** *The effect of labor flexibility (numerical: H2a, functional: H2b, financial: H2c, time: H2d) on net profit will be strengthened by the labor relations climate.*

### 3. Methods

*3.1. Data Collection and Sample Characteristics*

This study is based on the Korea Labor Institute Workplace Panel Survey conducted in 2012. Korea's economic growth rate has continued to decline since 2012, and according to a report by the WEF, there was no significant change in labor flexibility during this period. Therefore, it can be seen that the data for 2012 is still valid. We plan to utilize the latest data in future research. A cross-sectional analysis was inevitable because there was insufficient firm data studied longitudinally. The 2012 survey covered 1770 workplaces, and financial data of 1489 firms were provided. The hypotheses were tested, as shown in Table 1, using a total of 1482 samples, excluding insincere responses.

*3.2. Definitions and Measurement of Variables*

This study categorized labor flexibility into four types (i.e., numerical flexibility, functional flexibility, financial flexibility, and time flexibility) based on the general classification system of labor flexibility used by many researchers, including Casey et al. [41] and Kornelakis [13], to first examine whether the firm in question uses labor flexibilization strategies.

3.2.1. Numerical Flexibility

This study used the definition of employment flexibility at the corporate level as the traditional numerical flexibility [13]. The use of various temporary workers and external workforce [42] was measured. Numerical flexibility was estimated by obtaining the ratio

of the sum of the number of temporary workers and outsourced workers out of the total population and then taking the logarithm of that ratio.

**Table 1.** Respondents of the Workplace Panel Survey.

| Category | Division | Number of Samples | % |
|---|---|---|---|
| Firm age | 63–113 years | 40 | 2.69 |
| | 13–62 years | 1123 | 75.77 |
| | 1–12 years | 319 | 21.52 |
| Firm size | Fewer than 50 persons | 296 | 19.97 |
| | 50–99 persons | 284 | 19.16 |
| | 100–199 persons | 292 | 19.70 |
| | 200–499 persons | 325 | 21.92 |
| | 500 persons or more | 285 | 19.23 |
| Industrial classification | Manufacturing | 685 | 46.22 |
| | Electricity, gas, and water supply | 23 | 1.55 |
| | Sewerage, waste management, materials recovery, and remediation activities | 11 | 0.74 |
| | Construction | 89 | 6.01 |
| | Wholesale and retail trade | 106 | 7.15 |
| | Transportation | 124 | 8.37 |
| | Hotels and restaurants | 28 | 1.89 |
| | Publishing, video, broadcasting, communications, and information services | 59 | 3.98 |
| | Financial and insurance activities | 57 | 3.85 |
| | Real estate activities, renting, and leasing | 8 | 0.54 |
| | Professional, scientific, and technical activities | 97 | 6.55 |
| | Business facilities management and business support services | 67 | 4.52 |
| | Public administration, defense, and social security | 6 | 0.40 |
| | Education | 24 | 1.62 |
| | Health and social work | 61 | 4.12 |
| | Arts, sports, and recreation-related services | 18 | 1.21 |
| | Membership organizations, repair, and other personal services | 19 | 1.28 |
| Labor union | Nonunion | 919 | 62.01 |
| | Union (single) | 508 | 34.28 |
| | Union (multiple) | 55 | 3.72 |
| Corporate form | Business corporations | 1253 | 84.55 |
| | Nonbusiness corporations (individual workplaces, school corporations, medical corporations, religious corporations, etc.) | 229 | 15.45 |
| Net profit | 0 or less | 282 | 18.97 |
| | 1 million—less than 100 million KRW | 108 | 7.30 |
| | 100 million—less than 1 billion KRW | 381 | 25.72 |
| | 1 billion—less than 10 billion KRW | 386 | 26.07 |
| | 10 billion—less than 100 billion KRW | 200 | 13.50 |
| | 100 billion KRW— | 97 | 6.55 |
| | 1 trillion KRW— | 28 | 1.89 |

### 3.2.2. Functional Flexibility

Functional flexibility is defined as the readjustment of internal staff and job details, thereby adjusting employment without reducing the number of workers [2]. To determine the utilization of functional flexibility, two survey items were used: "Does your workplace have a regular and deliberate job rotation intended for multi-functionalization or acquisition of diverse work experiences?" and "Does your workplace officially provide multifunctional training?" The score was 1 when any of the two—regular job rotation and multifunctional training—were implemented, and 0 when neither was executed.

### 3.2.3. Financial Flexibility

Financial flexibility or pay flexibility (i.e., wage flexibility) is flexibility related to wages. This type signifies a shift from the rigid wage system determined in the past by seniority

or group negotiations to a performance-related pay system aligned with individual team-based abilities and performance, or implementing the productivity-based wage system, such as the wage peak system. The score was 1 when any of the two—performance distribution or wage peak system—were implemented, and 0 when neither was applied.

### 3.2.4. Time Flexibility

Time flexibility is the case wherein the employer adjusts working hours without adjusting the number of workers within the company. In this study, time flexibility was measured based on whether selective and flexible working hours were adopted, such as "Does your workplace operate selective working hours?" and "Does your workplace operate flexible working hours?" The score was 1 when any of the two—selective working hours or flexible working hours—were implemented, and 0 when neither was utilized.

### 3.2.5. Labor Relations Climate

This study concentrated on labor relations climate in terms of labor–management partnerships. The "overall labor relations climate" was considered for the following reasons. Studies on labor unions show jumbled empirical results because of the complexity of labor relations and on-site labor politics that are difficult to identify with just the existence of a labor union. By including the case wherein there is no labor union, it is possible to review the actual impact of labor relations separately from the organization and structure of labor unions. The item used to measure labor relations climate was "How are the overall labor relations of your workplace?" This item was rated on a five-point scale.

### 3.2.6. Corporate Performance

Out of all objective performance measurement indicators, net profit data was used to measure firms' financial performance. According to previous studies, the company's performance was measured as net profit [43].

### 3.2.7. Control Variables

Control variables that may affect net profit include firm age, firm size, labor-to-sales ratio, operating body, corporate form, and industry type, and the details of each variable are as follows. Firm age is the life span of each firm. The year of foundation was deducted from 2012, and a logarithm of that value was taken. Firm size was computed by utilizing the average number of workers in the workplace, calculated as the log of the average number of workers. The firm's labor-to-sales ratio was added as a control variable to control the cost, using the ratio dividing total labor costs by current revenues. Total labor costs include wage, retirement benefit, welfare benefit, and stock option. The operating body is a variable about whether the operator of the firm or business is a public institution or a private enterprise. Out of all the samples, 109 public institutions that participated in the survey were scored 1, and the rest in the private sector (private enterprises) were scored 0 as dummy variables. For corporate form, incorporated companies, such as corporations, limited liability companies, joint-stock limited partnerships, unlimited partnership companies, and foreign companies, were scored 1, and school corporates or medical corporations were scored 0. Industry type was classified into manufacturing and nonmanufacturing contingent on the industrial division system of the survey. Manufacturing was scored 1; nonmanufacturing was scored 0.

## 4. Results

This study analyzed the data using STATA 13.0 to verify the interaction effect among variables presented in the hypotheses. For hypothesis testing, Pearson's correlation and regression analyses were conducted, and hierarchical regression analysis was performed to ascertain the moderating effect of the overall labor relations climate. As shown Table 2, numerical flexibility relayed a positive correlation with time flexibility (0.07), and functional flexibility signified a high correlation with financial flexibility (0.13) and time flexibil-

ity (0.17). Additionally, the correlation coefficient of financial flexibility and time flexibility was 0.07, confirming a high correlation. Labor relations climate showed a positive correlation with functional flexibility (0.08), financial flexibility (0.10), and time flexibility (0.08).

**Table 2.** Correlations.

| Category | Mean | s.d | 1 | 2 | 3 | 4 | 5 | 6 | 7 | 8 | 9 | 10 | 11 |
|---|---|---|---|---|---|---|---|---|---|---|---|---|---|
| 1. Firm age | 3.06 | 0.62 | | | | | | | | | | | |
| 2. Firm size | 5.10 | 1.27 | 0.30 ** | | | | | | | | | | |
| 3. Labor-to-sales ratio | 0.25 | 0.29 | −0.10 ** | −0.01 ** | | | | | | | | | |
| 4. Operating body | 0.93 | 0.26 | 0.00 | −0.17 ** | 0.03 | | | | | | | | |
| 5. Corporate form | 0.85 | 0.36 | −0.02 | −0.08 ** | −0.19 ** | 0.52 ** | | | | | | | |
| 6. Industry type | 0.46 | 0.50 | 0.08 ** | −0.06 * | −0.36 ** | 0.25 ** | 0.27 ** | | | | | | |
| 7. Numerical flexibility | −1.97 | 1.41 | −0.11 ** | −0.07 * | 0.04 | −0.11 ** | −0.10 ** | −0.19 ** | | | | | |
| 8. Functional flexibility | 0.48 | 0.50 | 0.03 | 0.20 ** | −0.07 * | −0.19 ** | −0.13 ** | −0.07 ** | 0.05 | | | | |
| 9. Financial flexibility | 0.47 | 0.50 | −0.01 | 0.09 ** | −0.11 ** | −0.07 ** | 0.04 | 0.03 | 0.05 | 0.13 ** | | | |
| 10. Time flexibility | 0.14 | 0.35 | 0.01 | 0.16 ** | −0.00 | −0.25 ** | −0.17 ** | −0.19 ** | 0.07 * | 0.17 ** | 0.07 ** | | |
| 11. Labor relations climate | 3.72 | 0.67 | −0.00 | 0.02 | −0.03 | −0.02 | 0.05 | 0.03 | −0.01 | 0.08 ** | 0.10 ** | 0.08 ** | |
| 12. Net profit | 64,661.00 | 355,218.00 | 0.01 | 0.16 ** | −0.11 ** | 0.05 | 0.07 ** | −0.00 | 0.08 * | 0.06 * | 0.07 ** | 0.07 ** | 0.06 * |

\* $p < 0.05$ \*\* $p < 0.01$.

### *Results of Hypotheses Testing*

Regression analysis was performed to evaluate the relationship between labor flexibility and net profit and the moderating effect of labor relations climate. Table 3 provides the results of the regression analysis that measured the effect of each variable on the net profit of firms. Model 1 included control variables and independent variables, and Model 2 included moderating variables and interaction terms.

**Table 3.** Regression analysis results.

| | | Model 1 | | Model 2 | |
|---|---|---|---|---|---|
| | | **B** | **s.e** | **B** | **s.e** |
| | Constant | −249,150.30 | 92,530.99 | −273,162.00 | 89,481.80 |
| | Firm Age | −25,466.04 | 21,249.39 | −20,086.80 | 21,043.05 |
| | Firm Size | 56,518.12 ** | 11,186.43 | 54,598.72 ** | 11,081.39 |
| Control Variables | Labor-to-sales ratio | −160,296.70 ** | 49,533.23 | −152,926.00 ** | 48,972.55 |
| | Operating Body | 125,121.10 * | 54,568.23 | 135,846.90 * | 54,085.11 |
| | Corporate Form | 36,800.91 | 44,299.62 | 39,717.00 | 44,152.38 |
| | Industry Type | −11,458.68 | 30,261.39 | −16,198.50 | 30,016.23 |
| | Numerical Flexibility | 25,762.03 ** | 9204.47 | 25,127.85 ** | 9125.10 |
| Independent Variables | Functional Flexibility | 28,078.63 | 26,881.21 | 27,643.47 | 26,628.55 |
| | Financial Flexibility | 57,553.03 * | 26,026.53 | 55,969.09 * | 25,835.72 |
| | Time Flexibility | 59,951.22 | 36,345.18 | 47,845.69 | 36,399.11 |
| | Labor Relations Climate (A) | | | 39,063.84 * | 19,009.80 |
| | Numerical Flexibility × (A) | | | 38,969.61 ** | 13,665.03 |
| Moderating Variable | Functional Flexibility × (A) | | | −20,561.10 | 38,673.80 |
| | Financial Flexibility × (A) | | | 87,836.27 * | 37,608.42 |
| | Time Flexibility × (A) | | | 97,500.23 | 52,359.99 |
| | F | 6.959 ** | | 6.414 ** | |
| | R2 | 0.0784 | | 0.1058 | |
| | Adjusted R2 | 0.0671 | | 0.0893 | |
| | ΔR2 | | | 0.0274 | |

\* $p < 0.05$ \*\* $p < 0.01$.

As a result of analyzing the effect of labor flexibility on firms' net profit, it was found that numerical flexibility (b = 25,762.03, $p < 0.01$) and financial flexibility (b = 57,553.03, $p < 0.05$) had a significant effect on net profit. However, functional flexibility (b = 28078.63, $p = 0.297$) and time flexibility (b = 59,951.22, $p = 0.099$) were not statistically significant. Hence, Hypothesis 1a (numerical flexibility) and Hypothesis 1c (financial flexibility) were supported, and Hypothesis 1b (functional flexibility) and Hypothesis 1d (time flexibility) were rejected.

The moderating effect was assessed to examine how the effect of labor flexibility on net profit changes depending on the labor relations climate. The results are as shown in

Model 2 of Table 3. The $R^2$ increase of the model, including the overall labor relations climate as the moderating variable, was 0.27, inferring a statistically significant increase compared to the model without the moderating variable. The outcome verifies that there was a moderating effect in numerical flexibility and financial flexibility. In particular, the interaction term of numerical flexibility and labor relations climate was b = 38,969.61, $p < 0.01$, and the interaction term of financial flexibility and labor relations climate was b = 87,836.27, $p < 0.05$. For numerical flexibility and financial flexibility, the better the overall labor relations climate, the greater the effect on firms' net profit. Thus, both Hypotheses 2a and 2c were supported. Figure 2a,b show the plot with moderate effects of labor relations climate on main effect. Labor relations climate scores are high (+1 s.d) and low (−1 s.d), respectively.

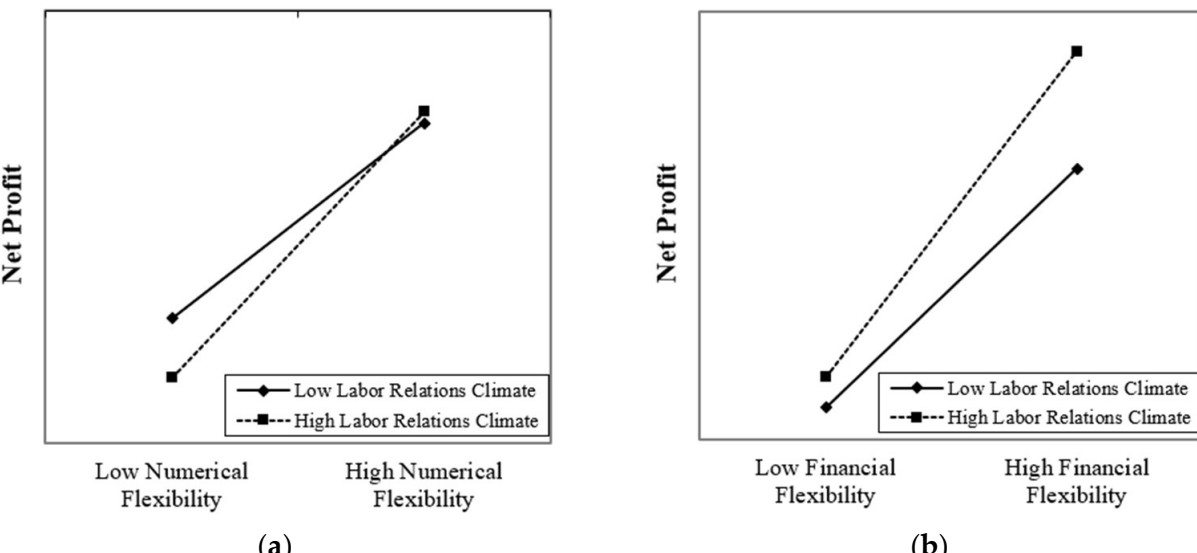

**Figure 2.** The Moderating Effect of Labor Relations Climate on Numerical and Financial Flexibility. (**a**) Numerical Flexibility and (**b**) Financial Flexibility.

## 5. Discussion

According to the results, numerical flexibility and financial flexibility have a significant positive effect on net profit, proving the need to maintain and expand the related practices. In addition, the positive effect of numerical and financial flexibility was strengthened as the overall labor relation climate was favorable. It was found that when the flexibility was used in an atmosphere of friendly labor relations, the net profit was improved. On the other hand, the functional and time flexibility variable did not have a meaningful statistical influence on the net profit. The reason why the positive effects of functional and time flexibility did not appear is that it takes time for functional and temporal flexibility to be exhibited. In Korean companies, workers are not often assigned to jobs that fit their duties, so even if they received multifunctional education, they would not have had an opportunity to demonstrate their capabilities.

### 5.1. Implications

As firms' business environment is rapidly changing, labor market flexibilization is becoming necessary to reinforce the competitiveness of Korean firms today. Therefore, this study tested the effects of four types of labor flexibility (i.e., numerical flexibility, functional flexibility, financial flexibility, and time flexibility) on firms' financial performance and investigated the factors that promote the effectiveness of labor flexibility.

Moreover, this study reviewed the moderating effect of the labor relations climate as a situational characteristic that impacts the actual effect of labor flexibility. Labor relations climate has a significant effect on how much firms' financial performance can be improved

by various labor flexibilization policies, systems, and techniques. By considering labor relations climate, which is a cognitive aspect of workers out of various attributes that represent labor relations, this research could more fundamentally and substantially analyze the effect of labor relations.

The findings demonstrate that numerical and financial flexibility had a statistically significant positive effect on performance, and these variables showed a stronger positive effect when the labor relations climate was better. These results were similar to previous studies [44,45]. In other words, using the aforementioned strategies within an amicable and positive labor relations climate considerably improves sustainable performance, which is represented by the firms' net profit.

On the contrary, the effect of functional and time flexibility on financial performance was not statistically significant. These results were different from previous studies [23,40]. Functional and time flexibility showed results that differed from other types of flexibility due to the time constraints and conditions to display multifunctional competencies. Because the Korean labor market still lacks a streamlined job analysis or job-based allocation, there are many cases wherein the distributed work does not completely match the workers' skills and competencies. Besides, there are considerable costs put into multifunctional training and regular job rotation, and a certain amount of time is needed by workers to internalize the multifunctionality that can facilitate financial performance.

The theoretical implications of this study are as follows. First, most of the previous literature constituted normative or policy studies on the effects of labor flexibility, with very few empirical studies. However, this study is significant because it verified the specific effects of labor flexibility through empirical research and confirmed its relationship with financial performance. Second, most studies conducted in Korea and overseas prioritized numerical flexibility and functional flexibility. Still, this study has significance because it examines all possible types of labor flexibility to empirically review the effect of each type. Third, this paper reviewed the situational factors necessary for firms to adopt various systems, making theoretical contributions by verifying the effect of labor relations climate between labor flexibility and financial performance and proving the necessity of labor relations climate as a moderating variable.

The practical implications of this study are as follows. First, this research validates that when firms use external human resources through numerical flexibility, they can flexibly cope with environmental changes and continuously generate competitive advantages. Firms must flexibly cope with the changes in the employment market by effectively using external human resources. Furthermore, government policies must be improved to support the efficient management of human resources by diversifying worker dispatch systems and forms of employment.

Second, this study provides important implications in that the business performance of firms can be improved by operating a performance distribution or wage peak system. When seeking wage flexibilization based on performance, financial flexibility must give a positive awareness to members with fair and reasonable rewards that are convincing and just. Hence, it is necessary to establish and implement extremely fair procedures and standards for financial flexibility, providing strict and reasonable evaluation methods that can convince the workers. Once workers feel that they are fairly evaluated, it will contribute significantly to the creation of a cooperative labor relations climate, which will cancel out various negative effects that may occur in financial flexibility.

Third, this paper implies that firms' hands-on workers must devote themselves to creating a labor relations climate. Managers must acknowledge workers as critical stakeholders instead of treating them as components of their firm, thereby reflecting their interests and systematically managing the labor relations climate from a long-term perspective. By creating an amicable and positive labor relations climate, workers can build a friendly image and trust toward the firm, perform their duties in a way that is desirable to the firm, engage in their work, and even more enthusiastically contribute to the betterment of performance.

*5.2. Limitations and Future Direction*

This paper has the following limitations and thus requires careful interpretation of the findings. First, this study used the 2012 data of the Workplace Panel Survey and thus has limitations with regards to cross-sectional data measurements. Hence, there may be an issue of reverse causality present because there is room for interpretation regarding labor flexibilization strategies that can be more easily introduced and implemented using the remaining resources as the firm has good sustainable performance.

Therefore, additional research must be conducted with time intervals using panel data analysis to overcome the limitations mentioned above. It is necessary to first review the long-term effects of labor flexibility after some time. For example, functional flexibility and time flexibility are related to learning, growth, participation, autonomy, and demands of workers, and thus they may show more long-term effects. Meanwhile, numerical flexibility and financial flexibility are related to control, regulations, cost reduction, and efficient performance, thereby showing more short-term effects. However, complaints may arise by hindering implicit knowledge accumulation among human resources due to frequent replacements or by reducing investment in human resources, which may increase conflicts within the organization. In this research, the positive effects set off the negative effects because only the short-term effects were reviewed. However, additional verification is mandatory to identify whether the negative effects can be set off in the long run, leaving only the positive effects.

Second, it is necessary to consider the interaction among the subfactors of labor flexibility. Considering that most firms will adopt two or more types of flexibility, there will be an interaction effect by adopting each type. Accordingly, the relationship among the types of flexibility, as well as their synergy and trade-off, must also be reviewed. Because a synergy can be created when one practice reinforces another [46], the effects may vary depending on whether only one type is used, or whether multiple or all four types are used. Thus, it is necessary to distinguish overall use from partial use, and also to categorize the four types into two similar dimensions and examine the relationship between the two, in tandem with the relationship between human resource performance and financial performance.

Third, there were certain constraints on converting the concepts into manipulated variables because this study used collected data instead of a survey created by the researcher. Specifically, variables of a labor relations climate could not be formed according to the previous research, and thus only "overall labor relations climate" was used in the measurement. The limitation of this study is that it had to rely only on the perception of respondents without subdividing the attributes. Furthermore, items on functional flexibility, financial flexibility, and time flexibility were measured based only on whether the relevant system was adopted or not. To overcome these limitations, more specific and detailed measurements on labor relations climate are needed.

Finally, financial performance in this study was measured with net profit, but it is necessary to add various other financial variables on costs and benefits, including nonfinancial performance variables. There must also be some research that verifies the effect of perception at the individual worker level to distinguish whether these psychological variables had a positive effect.

This study examined labor flexibility and financial performance of firms and the moderating effect of a labor relations climate. Firms are greatly influenced by the changes in the business environment, as well as technological development. Therefore, it is necessary to break free from the dichotomous logic wherein there is a conflict between employment security and labor flexibility in order to achieve worker satisfaction, employment security, generation of corporate profits, and flexibility increase. In this aspect, in-depth empirical research must be conducted on "flexicurity," which amalgamates both labor flexibility and employment security. Labor flexibility models that overcome the conventional pattern of conflicts can only be created after sufficiently discussing and agreeing on how to achieve both security and flexibility.

**Author Contributions:** Methodology, Y.K. and S.M.; Resources, Y.K.; Writing—original draft, H.C.; Writing—review & editing, S.M. All authors have read and agreed to the published version of the manuscript.

**Funding:** This research received no external funding.

**Institutional Review Board Statement:** Not applicable.

**Informed Consent Statement:** Not applicable.

**Data Availability Statement:** Not applicable.

**Acknowledgments:** This study utilized part of "A Study on the Development and Application of Diagnostic Tools for ex-Leadership Innovation" published by the Korea Expressway Corporation Research Institute in 2021.

**Conflicts of Interest:** The authors declare no conflict of interest.

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
