# Peer review of "The Effect of Labor Flexibility on Financial Performance in Korea: The Moderating Effect of Labor Relations Climate"

_sustainability, doi:10.3390/su14042121_

Round 1

Reviewer 1 Report

This paper set four hypotheses about the relationship between four types of flexibility and performance, but the there was no explanations about LR climate's moderating effect on each hypothesis. Since  H1a-H1d are based on all different reasons, moderation effects can also be based on different reasons. Indeed, some were supported and others are rejected after all. Thus, the hypotheses on moderation should include more specific explanations. 

Author Response

Point 1:  This paper set four hypotheses about the relationship between four types of flexibility and performance, but the there was no explanations about LR climate's moderating effect on each hypothesis. Since H1a-H1d are based on all different reasons, moderation effects can also be based on different reasons. Indeed, some were supported and others are rejected after all. Thus, the hypotheses on moderation should include more specific explanations. 

Response 1:
Thanks for the advice. We added next paragraph in the manuscript.

Numerical flexibility has a negative impact on the employment security (Barbieri, 2009). However, if the labor relation climate is favorable, workers will feel less job insecurity. A good labor relation climate will not only mitigate the negative effects, but the numerical flexibility agreed upon between labor and management will make the company expect it to be a driving force to get out of crisis or achieve better performance. Therefore, labor relation climate will positively moderate the relationship between numerical flexibility and financial performance.
Learning various tasks is an important career path that increases the employability of workers, but excessive flexibility causes overwork. (Salvador et al., 2020). However, if the labor relation climate is favorable, workers can develop their careers and expect better performance and rewards through the functional flexibility. Therefore, labor relation climate will positively moderate the relationship between functional flexibility and financial performance.
The positive effect of the pay-for-performance system on the motivation of employees is reinforced when labor-management relations are good. This is because, in the case of pay-for-performance, fair evaluation must precede, and if labor relation climate is favorable, employees will trust that the performance evaluation is done fairly. Therefore, labor relation climate will positively moderate the relationship between financial flexibility and financial performance. 
Time flexibility is related to work autonomy (Lott, 2015). When the labor relation climate is favorable, the motivation and commitment will increase because the given autonomy is genuinely felt. In addition, workers will feel that support and treatment for them will increase, so they will become more commit in their work. Therefore, labor relation climate will positively moderate the relationship between time flexibility and financial performance.

Reviewer 2 Report

Dear authors,
Your idea is important and interesting; however, several aspects must be addressed before it could be consider for publication:
a)    As your work is highly context dependent, you must present more details about Korean economy and business environment; moreover, you must make clear why labor flexibility is a problem in the Korean context, so others could understand and make connections to theirs “context”; 
b)    Using business performance to label net profit is misleading. Please declare your hypotheses using the variable correctly. I have appreciated your limitations section about this variable, but you have neglected the potential of using of the control variables results from both models to better support your results.
c)    If I understand correctly your data refers to 2012? You should explain why this data reflects Korean today's business context;
d)    You have aggregated data from different industrial classification, different firm sizes. This kind of aggregation brings positive and negative aspects to the research. However, you do not explicitly address to those aspects and how it may affect your results. As an example you could present and discuss results for each sub group.
e)    About table 3 the columns need a heading.
f)    Your discussion should be review. You must insert yours discovers into the context and use the results from the control variables.
g)    The literature reviewed must be update, please note that only 11% of your references list have less than 7 years.
h)    Finally, you are submitting your work to a journal that “is an international, cross-disciplinary, scholarly, peer-reviewed and open access journal of environmental, cultural, economic, and social sustainability of human beings. It provides an advanced forum for studies related to sustainability and sustainable development”, and I couldn’t find a single paragraph addressing sustainability and how your work contributes to improve it in the business context.

Author Response

Point 1: 
As your work is highly context dependent, you must present more details about Korean economy and business environment; moreover, you must make clear why labor flexibility is a problem in the Korean context, so others could understand and make connections to theirs “context”; 

Response 1: 
Thanks for the advice. We added next paragraph in the manuscript.

Korea's economic growth rate has continuously declined since 2010, and the growth rate in 2020 is -0.9%, showing a low growth rate. In addition, the economic uncertainty that companies are facing has not been resolved. According to the report of the World Economic Forum (WEF), Korea's labor market rigidity ranks 76th, which is 7 steps lower than last year. The reason that labor flexibility is important is that effective use of labor force is necessary for the sustainable survival of an organization. 

Point 2:
Using business performance to label net profit is misleading. Please declare your hypotheses using the variable correctly. I have appreciated your limitations section about this variable, but you have neglected the potential of using of the control variables results from both models to better support your results.

Response 2:
We replaced "business performance" with "net profit" following the reviewer's advice.

Point 3:
If I understand correctly your data refers to 2012? You should explain why this data reflects Korean today's business context;

Response 3:
Korea's economic growth rate has continued to decline since 2012, and according to a report by the WEF, there was no significant change in labor flexibility during this period. Therefore, it can be seen that the data for 2012 is still valid. We plan to utilize the latest data in future research.

Ponit 4:
You have aggregated data from different industrial classification, different firm sizes. This kind of aggregation brings positive and negative aspects to the research. However, you do not explicitly address to those aspects and how it may affect your results. As an example you could present and discuss results for each sub group.

Response 4:
Thanks for the good point. We included "firm age, firm size, labor-to-sales ratio, operating body, corporate form, and industry type" as control variables to control for effects reported by reviewers. Although not reported in this paper, there was little difference between subgroups. 

Point 5:
About table 3 the columns need a heading.

Response 5:
We added titles to Columns. thank you.

Ponit 6:
Your discussion should be review. You must insert yours discovers into the context and use the results from the control variables.

Response 6:
We added next paragraph in manuscript.
According to the analysis results, numerical flexibility and financial flexibility have a significant positive effect on net profit, proving the need to maintain and expand the related practices. In addition, the positive effect of numerical and financial flexibility was strengthened as the overall labor relation climate was favorable. It was found that when the flexibility was used in an atmosphere of friendly labor relations, the net profit was improved. On the other hand, the functional and time flexibility variable did not have a meaningful statistical influence on the net profit. The reason why the positive effects of functional and time flexibility did not appear is that it takes time for functional and temporal flexibility to be exhibited. In Korean companies, workers are not often assigned to jobs that fit their duties, so even if they received multi-functional education, they would not have had an opportunity to demonstrate their capabilities.

Point 7:
The literature reviewed must be update, please note that only 11% of your references list have less than 7 years.

Response 7:
We updated our references based on the reviewer's advice. The proportion of papers published after 2014 is 50%.

Point 8:
 Finally, you are submitting your work to a journal that “is an international, cross-disciplinary, scholarly, peer-reviewed and open access journal of environmental, cultural, economic, and social sustainability of human beings. It provides an advanced forum for studies related to sustainability and sustainable development”, and I couldn’t find a single paragraph addressing sustainability and how your work contributes to improve it in the business context.

Response 8:
In this study, it is assumed that sustainable growth is achieved through labor flexibility. However, we did not clearly write it. We followed the reviewer's advice and clearly stated that it was a sustainable performance throughout the manuscript. Thank you for your kind comments. 

Reviewer 3 Report

This is a very interesting paper, and it contributes to theory by adding labour relations to the theory of flexibility.

There are some shortcomings that needs to be addressed before publication. First, the aim of the paper should be better stated in the introduction. Second, it is needed to give argument for using profit as an indicator of firm performance instead of other indicators. Third, using data from 2012 needs to be addressed. Are newer data available. What has happen in Korea since 2012? Is it likely that the results are still valid? Please address in the discussion section. Forth, please give some indication in the discussion section why functional and time flexibility did not have an effect on profit. Finally, please relate your findings to former research in the discussion section.

Author Response

There are some shortcomings that needs to be addressed before publication. 
Point 1:
First, the aim of the paper should be better stated in the introduction. 

Response 1:
Thank you for comments. In this study, we will classify the sub-constructs of labor flexibility and examine the effect of each type of labor flexibility on net profit. Also, from a contingency perspective, this study aims to demonstrate the effect of the labor relations climate moderating the relationship between labor flexibility and net profit. 

Point 2:
Second, it is needed to give argument for using profit as an indicator of firm performance instead of other indicators. 

Response 2:
Thank you for comments. We added next sentence. 
According to previous studies, the company's performance was measured as net profit[42]. 

Point 3:
Third, using data from 2012 needs to be addressed. Are newer data available. What has happened in Korea since 2012? Is it likely that the results are still valid? Please address in the discussion section. 

Response 3:
Korea's economic growth rate has continued to decline since 2012, and according to a report by the WEF, there was no significant change in labor flexibility during this period. Therefore, it can be seen that the data for 2012 is still valid. These issues will be additionally described in the limitation, and we plan to utilize the latest data in future research.

Point 4:
Forth, please give some indication in the discussion section why functional and time flexibility did not have an effect on profit. 

Response 4:
We added next paragraph in manuscript.
According to the analysis results, numerical flexibility and financial flexibility have a significant positive effect on net profit, proving the need to maintain and expand the related practices. In addition, the positive effect of numerical and financial flexibility was strengthened as the overall labor relation climate was favorable. It was found that when the flexibility was used in an atmosphere of friendly labor relations, the net profit was improved. On the other hand, the functional and time flexibility variable did not have a meaningful statistical influence on the net profit. The reason why the positive effects of functional and time flexibility did not appear is that it takes time for functional and temporal flexibility to be exhibited. In Korean companies, workers are not often assigned to jobs that fit their duties, so even if they received multi-functional education, they would not have had an opportunity to demonstrate their capabilities. 

Point 5:
Finally, please relate your findings to former research in the discussion section.

Response 5:

We revised the next paragraph in the discussion section. 

The findings demonstrate that numerical and financial flexibility had a statistically significant positive effect on performance, and these variables showed a stronger positive effect when the labor relations climate was better. These results were similar to previous studies[43,44]. In other words, using the aforementioned strategies within an amicable and positive labor relations climate considerably improves sustainable performance, which is represented by the firms’ net profit. 
On the contrary, the effect of functional and time flexibility on business performance was not statistically significant. These results were different from previous studies[22,39]. Functional and time flexibility showed results that differed from other types of flexibility due to the time constraints and conditions to display multifunctional competencies. Since the Korean labor market still lacks a streamlined job analysis or job-based allocation, there are many cases wherein the distributed work does not completely match the workers’ skills and competencies. Besides, there are considerable costs put into multifunctional training and regular job rotation, and a certain amount of time is needed by workers to internalize the multifunctionality that can facilitate business performance.

Round 2

Reviewer 1 Report

Well revised, but since line 195-220 were added, line 186-194 should also be revised accordingly. 

Author Response

Point 1:

Well revised, but since line 195-220 were added, line 186-194 should also be revised accordingly. 

Response 1:

Thank you for comments. Line186-194 has been deleted according to the reviewer's advice because the newly added paragraph contains same logic.

Reviewer 2 Report

Thank you for our work in improving the manuscript. 
However, your manuscript will the stronger if you integrate into the text the answers that you have provide for the points 3, 4.
About point 8, our text must clearly state why and how labor flexibility is a way to achieve business sustainability. 

Author Response

Point 1:

However, your manuscript will the stronger if you integrate into the text the answers that you have provide for the points 3, 4.

Response 1:

Thank you for comments. We included next paragraph in the manuscript. 
“Korea's economic growth rate has continued to decline since 2012, and according to a report by the World Economic Forum, there was no significant change in labor flexibility during this period. Therefore, it can be seen that the data for 2012 is still valid. We plan to utilize the latest data in future research.”

Point 2:

About point 8, our text must clearly state why and how labor flexibility is a way to achieve business sustainability. 

Thank you for comments. We included next paragraph in the manuscript(introduction).

“Labor flexibility is necessary to maintain competitiveness and achieve sustainable growth in a rapidly changing environment. This is because, in order to satisfy the rapidly changing needs of customers, labor flexibility is effective, and if the labor flexibility can be maintained continuously, the competitive advantage is sustainable.”